# Knowledge Graph Finetuning Enhances Knowledge Manipulation in Large Language Models

**Hanzhu Chen[1], Xu Shen[2], Jie Wang[1]\*, Zehao Wang[1], Qitan Lv[1], Junjie He[1], Rong Wu[3], Feng Wu[1], Jieping Ye[2]**

[1] MoE Key Laboratory of Brain-inspired Intelligent Perception and Cognition,
University of Science and Technology of China
`{chenhz, zh-wang, qitanlv, hjunjie}@mail.ustc.edu.cn`
`{jiewangx,fengwu}@ustc.edu.cn`
[2] Independent Researcher
`{shenxuustc, jieping}@gmail.com`
[3] Zhejiang University
`{wurong1159}@zju.edu.cn`

## Abstract

Despite the impressive performance of general large language models(LLMs), many of their applications in specific domains (e.g., low-data and knowledge-intensive) still confront significant challenges. Supervised fine-tuning (SFT)—where a general LLM is further trained on a small labeled dataset to adapt for specific tasks or domains—has shown great power for developing domain-specific LLMs. However, existing SFT data primarily consist of Question and Answer (Q&A) pairs, which poses a significant challenge for LLMs to comprehend the correlation and logic of knowledge underlying the Q&A. To address this challenge, we propose a conceptually flexible and general framework to boost SFT, namely **K**nowledge **G**raph-Driven **S**upervised **F**ine-**T**uning (**KG-SFT**). The key idea of KG-SFT is to generate high-quality explanations for each Q&A pair via a structured knowledge graph to enhance the knowledge comprehension and manipulation of LLMs. Specifically, KG-SFT consists of three components: *Extractor*, *Generator*, and *Detector*. For a given Q&A pair, (i) *Extractor* first identifies entities within Q&A pairs and extracts relevant reasoning subgraphs from external KGs, (ii) *Generator* then produces corresponding fluent explanations utilizing these reasoning subgraphs, and (iii) finally, *Detector* performs sentence-level knowledge conflicts detection on these explanations to guarantee the reliability. KG-SFT focuses on generating high-quality explanations to improve the quality of the Q&A pair, which reveals a promising direction for supplementing existing data augmentation methods. Extensive experiments on **fifteen** different domains and **six** different languages demonstrate the effectiveness of KG-SFT, leading to an accuracy improvement of up to 18.1% and an average of 8.7% in low-data scenarios.

## 1 Introduction

Large language models (LLMs), such as GPT-4(Achiam et al., 2023), LlaMA 3(Touvron et al., 2023a), and Claude 3 (cla), have exhibited remarkable power and impressive versatility across a wide range of domains (Zhao et al., 2021; Brown et al., 2020; El-Kassas et al., 2021). However, applying LLMs to low-data and knowledge-intensive domains (e.g., a specific medical field (Nori et al., 2023) or private data with niche protocols (Cui et al., 2023; Li et al., 2023)) remain still challenging.

Recently, extensive research efforts have been devoted to boosting general LLMs performance in particular domains. One innovative training paradigm, Supervised Fined-Tuning (SFT), has emerged as a new trend and shown superior performance to enhance capabilities and controllability of general LLMs in certain domains (Zhang et al., 2023). The key idea of SFT is to adapt pre-trained LLMs to a specific task by continuing the training process on a labeled dataset, which

---
\*Corresponding author.

allows the model to refine its parameters for enhanced performance on task-relevant features (Wei et al., 2021). However, for certain domains, off-the-shelf SFT data in knowledge-intensive and low-data domains is generally scarce, and the process of creating high-quality SFT data necessitates considerable human effort and expertise, limiting the wide application of domain LLMs construction (Li et al., 2024a). Canonical methods to enrich the quantity of Q&A in SFT data and enhance LLMs performance are data augmentation. Traditional natural language processing methods such as easy-data-augmentation (EDA) including synonym replacement, character replacement, random swapping, and back translation (Wei & Zou, 2019; Belinkov & Bisk, 2017; Coulombe, 2018; Wang et al., 2022). Recently, several endeavors have explored using an LLM to expand the SFT dataset. AugGPT (Dai et al., 2023) utilizes an LLM (such as ChatGPT) to rephrase questions. GPT3Mix (Yoo et al., 2021) enhances SFT data by prompting an LLM to generate similar questions to those in the SFT data through few-shot prompts.

Despite the effectiveness of these augmentation methods in scaling up the quantity of SFT data, the vanilla SFT data augmentation method still confronts a significant challenge that may hinder the domain-specific fine-tuning of LLMs—the lack of correlation and logic between the knowledge underlying the SFT data. Existing SFT data are mainly structured merely in the form of Q&A, whereby LLMs during the SFT process simply acquire the superficial patterns (such as the output space and format) of Q&A (Kung & Peng, 2023) and do not comprehend the correlation and logic of knowledge underlying the Q&A pairs. For example, for the question: *Which is not a common symptom of cancer, persistent fever, or weight gain?* The answer: *Weight gain*. This involves multiple pieces of knowledge, such as "Cancer can cause a decrease in the body's resistance", "A decrease in resistance usually causes persistent fever", "Cancer cells consume a large amount of energy", and "Energy consumption can lead to weight loss". This fragmented knowledge in pre-training makes it difficult for LLMs to recall relevant knowledge for logical reasoning when answering questions. As a result, even after undergoing substantial training with sufficient SFT data, fine-tuned LLMs still cannot effectively manipulate the knowledge within the pre-training data, specifically in terms of recall, reasoning, and transfer (Zhu & Li, 2023; Allen-Zhu & Li, 2023).

Therefore, in this paper, we seek to answer the question: *Can we not only focus on just augmenting the **quantity** but also the **quality** of the SFT training data, i.e., revealing the correlation and logic of knowledge underlying the SFT data?* With the previous Q&A pair as an example, it involves the correlation and logic of knowledge as follows: "*cancer–may cause–>decreased resistance–may cause–>persistent fever*", and "*cancer cells–may cause–>energy consumption–may cause–>weight loss*". This corresponds well to the content within a series of triples (i.e., subgraphs) in the knowledge graph (KG). We explore the introduction of KGs to generate high-quality explanations to promote better comprehension for each Q&A pair. Thus, we propose a novel approach, namely **K**nowledge **G**raph-Enhanced **S**upervised **F**ine-**T**uning (**KG-SFT**), which can elucidate the correlation and logic of knowledge to enhance the knowledge manipulation (e.g., knowledge recall, reasoning, and transfer) ability of LLMs.

KG-SFT is a novel framework and effectively generates explanations that are logical, fluent, and trustworthy. Specifically, these three characteristics are aligned with the three components of KG-SFT.

(i) *Extractor* integrates external open-source knowledge graphs such as UMLS (Bodenreider, 2004) to identify entities within Q&A pairs. *Extractor* also retrieves their multi-hop reasoning subgraph between them to reveal the correlation and logic of knowledge underlying the Q&A pairs.

(ii) *Generator* uses a graph-structure significance scoring algorithm, HIST (Kleinberg, 1999), to score entities and relations within the reasoning subgraph. *Generator* selects the higher-scoring parts as the significant content for LLMs to let the LLMs generate a fluent draft explanation to the Q&A pairs.

(iii) *Detector* splits the draft explanations at the sentence level and detects the potential knowledge conflicts with the reasoning subgraph. *Detector* also reprompts to regenerate the conflict explanations.

Extensive experiments on **fifteen** different domains and **six** different languages demonstrate the effectiveness of KG-SFT, leading to a maximum accuracy improvement of up to 18.1% and an average of 8.7% in low-data scenarios. Indeed, given the significant emphasis on accuracy in many practical low-data domains, an average improvement of 8.7% may represent substantial economic potential (Hendrix et al., 2022; Wolff et al., 2020). We also conduct knowledge manipulation

experiments to evaluate the model's advancements in recall, reasoning, and transfer capabilities. KG-SFT can also be an effective plug-and-play module to incorporate with quantity augmenting methods.

## 2 RELATED WORK

### 2.1 TEXT DATA AUGMENTATION

Data augmentation is a classical research area in natural language processing. Traditional data augmentation techniques primarily focus on character and word-level enhancements. For example, EDA (Wei & Zou, 2019) utilizes random insertion, random swapping, random deletion, and synonym replacement to enrich data diversity(Belinkov & Bisk, 2017; Coulombe, 2018; Wang et al., 2022). Recently, techniques based on language models have enabled sentence or even document-level augmentation, with methods based on cutting-edge LLMs demonstrating powerful competitive advantages (Deng et al., 2023; Fang et al., 2023; Ubani et al., 2023). A noteworthy example is AugGPT (Dai et al., 2023), which utilizes an LLM (such as ChatGPT) to rephrase questions in SFT data to diversify the expression forms of Q&A. Moreover, GPT3Mix (Yoo et al., 2021) enhances SFT data by prompting an LLM to generate similar questions to those in the SFT data through few-shot prompts.

### 2.2 KNOWLEDGE GRAPH ENHANCED LLMS

Knowledge graphs (KGs) are considered a promising technology for addressing the limitations of large language models (LLMs) in terms of inference and interpretability, given their advantages in structured knowledge representation (Pan et al., 2024). Recent research has mainly focused on converting structured knowledge from KGs into textual prompts to enhance the question-answering capabilities of LLMs (Chen et al., 2024; Lv et al., 2024). For example, Think-on-Graph (ToG) (Sun et al., 2023) utilizes iterative beam search on a KG to improve reasoning; KGR (Guan et al., 2024) autonomously retrofits LLM responses with validated factual statements from KGs; and KAPING (Baek et al., 2023) enhances zero-shot question answering by appending retrieved facts from KGs to LLM inputs. Retrieval-augmented methods primarily provide factual knowledge to LLMs during the reasoning phase. In contrast, our KG-SFT focuses on elucidating the correlation and logic between knowledge by generating high-quality training data, thereby significantly enhancing the knowledge manipulation capabilities of LLMs.

## 3 PRELIMINARIES

### 3.1 BM25 ALGORITHM

For a given document $d$ and a query $q$ containing keywords $q_1, q_2, ..., q_n$, the BM25 score of $d$ with respect to $q$ is computed as follows:$\text{BM25}(d, q) = \sum_{i=1}^{n} \text{IDF}(q_i) \cdot \frac{f(q_i, d) \cdot (k_1 + 1)}{f(q_i, d) + k_1 \cdot (1 - b + b \cdot \frac{\text{len}(d)}{\text{avgdl}})}$, where $f(q_i, d)$ is the term frequency of $q_i$ in $d$, $\text{len}(d)$ is the length of the document $d$ (in words), avgdl is the average document length in the text collection from which documents are drawn, $k_1$ and $b$ are free parameters usually chosen, without loss of generality, as $k_1 = 1.2$ to $2.0$ and $b = 0.75$, and $\text{IDF}(q_i)$ is the inverse document frequency of $q_i$ across the collection of documents, defined as: $\text{IDF}(q_i) = \log \frac{N - n(q_i) + 0.5}{n(q_i) + 0.5}$, where $N$ is the total number of documents in the collection and $n(q_i)$ is the number of documents containing $q_i$.

### 3.2 HITS ALGORITHM

The Hyperlink-Induced Topic Search (HITS) (Kleinberg, 1999), also known as Hubs and Authorities, is an algorithm used to rate web pages. As for knowledge graphs, entities can be viewed as pages, where a hub is an entity that points to many other entities (authorities), and authority is an entity that is pointed to by many hubs. The iterative algorithm updates the hub and authority scores for each entity based on its relationships, with the key equations being: $h(e_i) = \sum_{e_j \in O(e_i)} a(e_j)$ and $a(e_i) = \sum_{e_j \in I(e_i)} h(e_j)$, where $h(e_i)$ and $a(e_i)$ are the hub and authority scores of entity $e_i$,

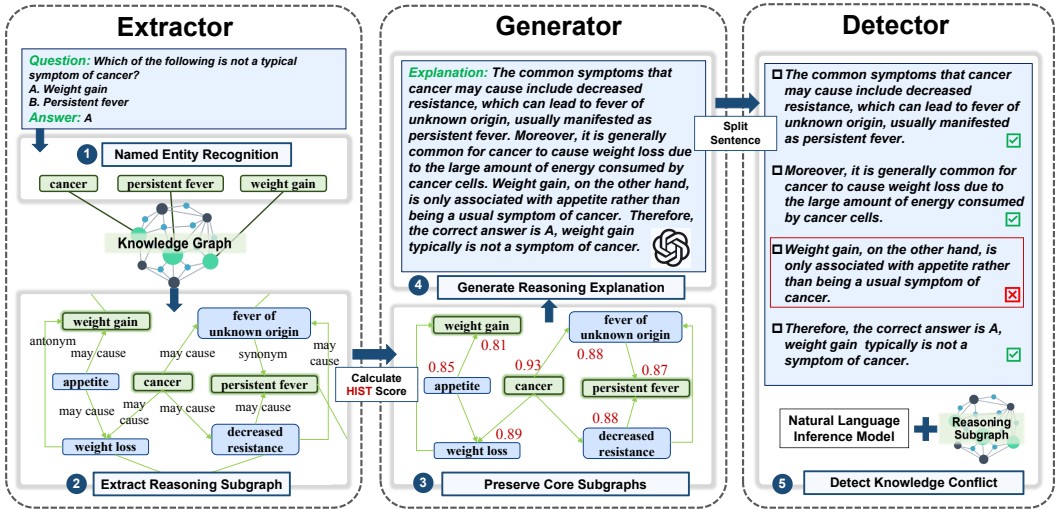

Figure 1: An overview of KG-SFT. KG-SFT integrates *Extractor*, *Generator*, and *Detector* to enhance the quality of vanilla SFT data. The workflow is as follows. (1) Perform Named Entity Recognition on the Q&A pair to extract potential entity list of question, options, and answer respectively. (2) Search the neighboring entities for each entity to obtain the reasoning subgraph. (3) Preserve core subgraphs that are strongly related to the Q&A pair via the HITS algorithm. (4) Generate reasoning explanation via an external LLM (5) Detect knowledge conflict via the Natural language inference model and the reasoning subgraph.

respectively, $O(e_i)$ is the set of entities that $e_i$ points to (out-links), and $I(e_i)$ is the set of entities that point to $e_i$ (in-links). The scores are normalized over all entities after each iteration. We refer to the mean of the final authority and hub score as the HIST score.

## 4 METHOD

We propose the KG-SFT framework to enhance the quality of the SFT data by revealing their underlying correlation and logic of knowledge. Specifically, KG-SFT consists of three components: *Extractor*, *Generator*, and *Detector*. An overview of KG-SFT is shown in Figure 1.

### 4.1 EXTRACTOR

*Extractor* first derives relevant reasoning subgraphs in the knowledge graph based on the Q&A pair to reveal the underlying correlation and logic of knowledge. Specifically, for a given Q&A pair, the workflow of *Extractor* is as follows:

(i) *Extractor* first conducts named entity recognition (NER) on the question, options, and answer to derive the entity list of question, options, and answer, respectively. Regarding the NER model, we employ the existing NER tools provided by the open-source knowledge graphs, specifically leveraging tools like Metamap from UMLS.

(ii) To mine the correlation between knowledge underlying the Q&A pair, after obtaining the list of entities, *Extractor* then enrich the neighbors of these entities within the external knowledge graph. We further apply the off-the-shelf BM25 (Robertson et al., 2009) algorithm to rank the triples (entity, relation, neighbor) based on their relevance to the Q&A text, retaining the top (default 20) related triples as candidates.

(iii) To mine the comprehensive logic between knowledge underlying the Q&A pair, *Extractor* finally retrieved three types of inference paths: from question entity to question entity, from option entity to option entity, and from question entity to answer entity.

By deduplicating and merging the triples obtained from the neighbor subgraph and inference path, we can derive a triple list to represent the reasoning subgraph. For a given Q&A pair, "Which of the following is not a typical symptom of cancer?" with options "A. Weight gain", "B. Persistent fever" and the correct answer is "B. Persistent fever". **First**, *Extractor* conducts NER to derive

the *entity list$_{question}$*= [Cancer], the *entity list$_{options}$*=[Weight gain, Persistent fever], and the *entity list$_{answer}$*=[Persistent fever]. **Then**, *Extractor* enriches the neighbors of these entities. For example, for "Cancer," *Extractor* enriches highly relevant triples such as (Cancer, May cause, Weight loss) and (Cancer, May cause, Fever of unknown origin). **Finally**, *Extractor* retrieves the inference paths, e.g., (Cancer, May cause, Decreased resistance) followed by (Decreased resistance, May cause, Persistent fever). The triples are finally combined to form the final list of triples for the reasoning subgraph.

## 4.2 GENERATOR

After extracting the reasoning subgraphs, *Generator* applies an LLM to create explanations for the given Q&A and transform the structured knowledge and logic underlying the questions into a natural language text format. *Generator* employs the off-the-shelf Hyperlink-Induced Topic Search (HITS) algorithm (Kleinberg, 1999) to filter the significant content within the reasoning subgraph.

Specifically, *Generator* **first** calculates the HIST scores of entities within the reasoning subgraphs via the HITS algorithm, which relies on the iterative updating of initial scores based on the structure of graphs. Note that to find content related to the Q&A, we will assign a higher initial score to the entity when it appears in the Q&A pair, while other entities will receive a lower score if the entity does not appear in the Q&A pair. **Then**, *Generator* selects the top (default 10) ranked neighbor subgraphs and inference paths by HIST scores as input to the LLMs (e.g. ChatGPT) to create draft explanations. The prompt used instructs the LLMs to generate clear explanations based on the provided question, answer, and triples. Please refer to Appendix A.4 for details.

Applying the HITS algorithm to the above-mentioned reasoning subgraph, we observe "Cancer" as an entity exhibiting high authority due to its close association with "May cause" across multiple central paths, notably highlighted through the relational chain of (Cancer, May cause, Fever of unknown origin). Meanwhile, "persistent fever," as the answer entity, achieves significant centrality through the path (Decreased resistance, May cause, Persistent fever).

Upon receiving these core triples, *Generator* produces draft explanations that reflect the logical relationship between the question and the answer: "The common symptoms that cancer may cause include decreased resistance, which can lead to fever of unknown origin, usually manifested as persistent fever. Moreover, it is generally common for cancer to cause weight loss due to the large amount of energy consumed by cancer cells. Weight gain, on the other hand, is only associated with appetite rather than being a usual symptom of cancer. Therefore, the correct answer is A, weight gain typically is not a symptom of cancer." Therefore, *Generator* conveys the underlying medical knowledge in a more fluent and clear manner.

## 4.3 DETECTOR

After generating draft explanations for each Q&A pair, *Detector* further examines these explanations using the triples from the inference graph to ensure their correctness. *Detector* aims to enhance the correctness of the generated explanations and minimize potential misguidance that may occur during the generation process by LLMs. Specifically, to generate the draft explanation, the detection process is as follows:

(i) Segment the draft explanation into sentences and then match them with the initially obtained entities list to form the matched comparison triples.

(ii) Input the matched comparison triples and segmented sentence explanations into an NLI model to assess for knowledge conflicts. Considering the input length and capability constraints of the NLI model, we directly input the comparison triples (five per group), combined with sentences, into an off-the-shelf state-of-the-art NLI model, DeBERTa (He et al., 2020; Xie et al., 2023) to determine the knowledge conflicts issue.

(iii) Mark a sentence with subsequent deletion tag, if it is detected with knowledge conflict. If an excessive number of sentences (default 30%) are found with knowledge conflicts, the re-prompt mechanism will re-guide the LLM to re-generate the explanations. The re-prompt instructs the model to reference the marked sentences containing knowledge conflicts and regenerate a new correct explanation. Please refer to Appendix A.4 for details.

For example, for "Weight gain, on the other hand, is only associated with appetite rather than being a usual symptom of cancer." combined with a triple (Appetite, May cause, weight gain) input into

DeBERTa, the probability of knowledge conflict obtained is greater than the predefined threshold and thus will be marked as a knowledge conflict.

# 5 EXPERIMENTS

To evaluate the effectiveness of our KG-SFT, we design a suite of experiments that not only demonstrate a significant enhancement in the SFT process for LLMs but also provide high-quality analytical experiments. To simulate a more realistic application scenario and prove the versatility of KG-SFT, we conduct experiments across six language settings: English, Chinese, French, Japanese, Russian, and Spanish. Specifically, we divide the experiments into *eight* parts:

- To comprehensively evaluate the enhancements of KG-SFT over the original SFT, we retain various proportions of the training set to simulate different scales of low-data scenarios.
- To demonstrate the superiority of KG-SFT, we conduct comparative experiments with existing baselines on datasets across six languages.
- To validate the potential of KG-SFT as a plug-and-play module, we conduct joint experiments on quantity and quality augmentation.
- To investigate the contribution of each component within KG-SFT, we conduct the ablation study of each component.
- To demonstrate the generalizability of KG-SFT, we incorporate over 10 diverse domain datasets from the multi-task language understanding benchmark.
- To further analyze why KG-SFT is effective, we conduct experiments on knowledge manipulation to explore the fine-tuned LLMs with KG-SFT.
    1. We explore the LLMs' knowledge recall ability by locating factual knowledge.
    2. We explore the LLMs' knowledge reasoning ability by multi-hop reasoning Q&A experiments.
    3. We explore the LLMs' knowledge transfer ability by multilingual transfer experiments, please refer to Appendix B.4 for details.
- To prove the generalizability of KG-SFT, we perform experiments on LLaMA-2-7B-chat (Touvron et al., 2023b), BLOOMZ-7B-chat (Muennighoff et al., 2022), and MMedLM2-7B (Qiu et al., 2024), please refer to Appendix B.5 for details.
- To investigate potential data leakage, we conduct a thorough analysis to ensure that the performance improvements of LLMs on the test set are not directly attributable to the KG content added to the training set, please refer to Appendix B.1 for details.

## 5.1 EXPERIMENT SETUPS

**Task and Datasets.** We choose the medical field as a canonical low-data and knowledge-intensive field, as high-quality supervised data is usually sparse, and medicine has rich and difficult factual knowledge. Moreover, evaluating LLMs conventionally relies on multiple-choice questions, which can provide an objective score (Qiu et al., 2024). Therefore, our evaluation task adopts multiple-choice questions and selects medical examination questions in six languages as the evaluation data. Please refer to Appendix A.1 for the statistics of our datasets.

**Models and Metrics.** Unless specified, we use LLaMA-2-7B-chat as the default backbone to evaluate our KG-SFT. We choose ChatGPT (gpt-3.5-turbo) and DeBERTa-v2 as our *Generator* and *Detector*. We use the accuracy rate of multiple-choice questions as metrics.

**Baseline Models.** We implement *twelve* variants of methods as our baselines for a comprehensive comparison. **(i) Vanilla:** standalone LLMs without any modification. **(ii) Vanilla SFT:** original supervised fine-tuning method based on Q&A dataset. **(iii) EDA-RS:** easy data augmentation by randomly removing words within sentences. **(iv) EDA-RS:** easy data augmentation by randomly swapping word positions within sentences. **(v) EDA-RI:** easy data augmentation by randomly inserting new words within sentences. **(vi) EDA-SR:** easy data augmentation by swapping words within sentences with their synonyms. **(vii) AugGPT:** utilizing an LLM (such as ChatGPT) to rephrase questions in SFT data to diversify the expression forms of Q&A. **(viii) GPT3Mix:** prompting an

LLM to generate similar questions to those in the SFT data through few-shot prompts. **(ix) CoT:** prompting an LLM to directly generate explanations based on Chain of Thought. Moreover, we introduce knowledge graph (KG)-enhanced methods, such as **(x) Think-on-Graph (ToG)**, which utilizes iterative beam search on a knowledge graph for improved reasoning; **(xi) KGR**, which autonomously retrofits LLM responses with validated factual statements from knowledge graphs; and **(xii) KAPING**, which enhances zero-shot question answering by appending retrieved facts from knowledge graphs to LLM inputs.

## 5.2 MAIN RESULTS

Table 1: Experiment results on the multiple-choice questions benchmarks in six languages range from different data ratios. For each dataset and data ratio, the numbers before/after the slash represent the accuracy rates for SFT and KG-SFT, respectively, with the **bold** indicating the best results.

| % Data | MedQA (English) | MedQA (Chinese) | IgakuQA (Russian) | RuMedDaNet (Spanish) | MedMCQA (French) | HeadQA (Japanese) |
|---|---|---|---|---|---|---|
| 5% | 26.02/**40.00** | 35.57/**38.83** | 21.80/**58.20** | 29.35/**36.49** | 12.90/**14.69** | 13.56/**16.58** |
| 10% | 39.89/**43.76** | 37.65/**43.63** | 42.57/**61.32** | 35.84/**40.66** | 13.56/**17.36** | 17.93/**19.90** |
| 20% | 43.04/**47.21** | 44.16/**47.70** | 46.88/**65.23** | 39.24/**42.37** | 16.12/**20.10** | 21.11/**21.60** |
| 50% | 44.61/**48.63** | 55.66/**57.85** | 53.12/**67.57** | 41.90/**44.71** | 21.73/**28.45** | 25.63/**28.14** |
| 100% | 47.80/**49.25** | 65.02/**67.86** | 65.62/**68.75** | 43.44/**46.49** | 27.37/**33.51** | 30.16/**32.66** |

In this section, we explore different data ratios to comprehensively evaluate the enhancements of KG-SFT over the original SFT. We set the data ratio from 5% to 100% to demonstrate the superiority of our KG-SFT in different augmented data scenarios. As shown in Table 1, KG-SFT achieves superior results across all data ratio and language settings over the vanilla SFT methods by a large margin. Notably, in the English scenarios, with only 5% of the augmented training data, KG-SFT leads to nearly 14% improvements over the vanilla methods. In the Russian scenarios, KG-SFT exhibits the most substantial performance gain at the 5% data ratio, from 21.8% to 58.20%. As the data ratio increases, KG-SFT still maintains superiority in all language scenarios as well.

It is worth noting that KG-SFT demonstrates superior performance across all languages, particularly in low-data scenarios. This highlights the effectiveness of generating high-quality explanations with corresponding the correlation and logic of knowledge underlying the Q&A pair. In high data ratio scenarios, although the improvement is limited, KG-SFT still maintains a performance lead across all language settings. This not only highlights the distinct advantages of KG-SFT when data availability is limited, but also indicates that KG-SFT can consistently enhance model performance, even in high-data scenarios.

## 5.3 RESULTS OF DIFFERENT BASELINES

First, as shown in Table 2, KG-SFT significantly outperforms these knowledge-enhanced methods / retrieval-augmented methods, such as TOG, KGR, and KAPING. This indicates that relying on simple retrieval-augmented methods may struggle to address the complexities of medical question-answering, as these questions often involve intricate knowledge and reasoning. Second, compared with the existing data augmentation baseline methods, KG-SFT achieves the optimal results across datasets in all six languages. Specifically, compared with traditional data augmentation methods such as random deletion, random swapping, random insertion, and synonym replacement, KG-SFT demonstrates higher performance scores across all languages. For instance, EDA-RD achieves an average score of 34.12, whereas KG-SFT shows an improvement of 7.67%. Furthermore, when compared with advanced data augmentation methods based on LLMs, such as AugGPT and GPT3Mix, KG-SFT still maintains its superior performance. An appealing feature of KG-SFT is that it generates high-quality explanations for each Q&A which enhances the correlation and logic of knowledge during the supervised fine-tuning process. These results demonstrate the effectiveness in real-world knowledge-intensive and low-data domains.

Table 2: Experiment results for vanilla LLM and different SFT variants. #Tuning QA refers to the final number of QA pairs for training enhanced by each method, with 1000 before augmentation. If the method does not require training, the #Tuning QA is "-". We **bold** the best results for each dataset.

| Method | #Tuning QA | MedQA (English) | MedQA (Chinese) | IgakuQA (Russian) | RuMedDaNet (Spanish) | MedMCQA (French) | HeadQA (Japanese) | Avg. |
|--------|-----------|---------|---------|---------|---------|---------|---------|------|
| Vanilla | - | 28.20 | 28.37 | 51.17 | 32.97 | 12.76 | 11.10 | 27.43 |
| TOG | - | 34.27 | 28.13 | 48.42 | 35.59 | 12.47 | 19.61 | 29.75 |
| KGR | - | 33.15 | 26.88 | 47.52 | 34.74 | 13.39 | 17.29 | 28.83 |
| KAPING | - | 36.39 | 27.24 | 54.66 | 34.98 | 11.54 | 15.91 | 30.12 |
| StructGPT | - | 35.16 | 24.50 | 55.32 | 36.16 | 14.24 | 20.01 | 30.90 |
| SFT | 1000 | 33.62 | 29.33 | 66.40 | 35.19 | 12.67 | 21.11 | 33.05 |
| EDA-RD | 2000 | 40.14 | 17.83 | 62.50 | 41.39 | 16.72 | 26.13 | 34.12 |
| EDA-RS | 2000 | 40.84 | 32.51 | 66.41 | 39.89 | 15.59 | 25.12 | 36.73 |
| EDA-RI | 2000 | 39.67 | 32.37 | 65.63 | 40.11 | 18.81 | 26.13 | 37.12 |
| EDA-SR | 2000 | 38.25 | 33.65 | 65.23 | 40.95 | 17.04 | 23.11 | 36.37 |
| AugGPT | 2000 | 40.29 | 36.54 | 62.14 | 40.70 | 22.99 | 27.13 | 38.30 |
| GPT3Mix | 2000 | 39.35 | 37.97 | 66.01 | 41.50 | 25.08 | 26.13 | 39.34 |
| CoT | 1000 | 37.65 | 39.01 | 65.23 | 40.33 | 25.08 | 23.63 | 38.49 |
| KG-SFT | 1000 | **41.71** | **39.31** | **68.75** | **44.40** | **28.45** | **28.14** | 41.79 |

Table 3: Experiment results for joint experiments on quantity and quality. We **bold** the best results for each comparative experiment. The row of MAX in the table is filled in with the best result for each dataset.

| Method | MedQA (English) | MedQA (Chinese) | IgakuQA (Russian) | RuMedDaNet (Spanish) | MedMCQA (French) | HeadQA (Japanese) |
|--------|---------|---------|---------|---------|---------|---------|
| AugGPT | 40.29 | 36.54 | 62.14 | 40.70 | 22.99 | 27.13 |
| AugGPT+KG-SFT | **40.92** | **40.45** | **68.35** | **43.14** | **27.33** | **28.63** |
| GPT3Mix | 39.35 | 37.97 | 66.01 | 41.50 | 25.08 | 26.13 |
| GPT3Mix+KG-SFT | **41.79** | **40.11** | **69.14** | **45.25** | **28.93** | **33.31** |
| EDA-RD | 40.14 | 17.83 | 62.5 | 41.39 | 16.72 | 26.13 |
| EDA-RD+KG-SFT | **41.39** | **37.62** | **69.92** | **43.18** | **27.81** | **28.14** |
| EDA-RS | 40.84 | 32.51 | 66.41 | 39.89 | 15.59 | 25.12 |
| EDA-RS+KG-SFT | **41.71** | **40.02** | **71.48** | **43.36** | **29.42** | **30.15** |
| EDA-RI | 39.67 | 32.37 | 65.63 | 40.11 | 18.81 | 26.13 |
| EDA-RI+KG-SFT | **41.24** | **38.29** | **67.18** | **42.26** | **29.58** | **33.16** |
| EDA-SR | 38.25 | 33.65 | 65.23 | 40.95 | 17.04 | 23.11 |
| EDA-SR+KG-SFT | **40.84** | **38.67** | **68.75** | **42.74** | **29.09** | **30.15** |
| KG-SFT | 41.71 | 39.31 | 68.75 | 44.40 | 28.45 | 28.14 |
| MAX | **41.79** | **40.45** | **71.48** | **45.25** | **29.42** | **33.31** |

## 5.4 Joint Experiments on Quantity and Quality

We conduct joint experiments on quantity and quality to demonstrate that KG-SFT can be incorporated with quantity-augmenting baselines as a plug-and-play module. As shown in Table 3, all quantity augmenting baselines achieve significant improvements by incorporating KG-SFT. For example, the accuracy of traditional EDA-RS in French increased from 15.59 to 29.42 with incorporating KG-SFT for quality enhancement, marking a relative improvement of 88.71% and even surpassing the original KG-SFT. Moreover, advanced baselines such as GPT3Mix achieve significant improvements by incorporating KG-SFT, outperforming the original KG-SFT in all values. These results highlight the significant potential when combining quantity augmenting methods with KG-SFT.

## 5.5 Ablation Study

To further investigate the contribution of each component within KG-SFT, we conduct a series of ablation experiments on the KG-SFT entire framework. Specifically, We denote KG-SFT without *Extractor*, i.e., without the knowledge graph, the LLM directly generates explanations, as KG-

Table 4: Results of ablation study on multi-Q&A datasets on all six languages, using LLaMA-2-7B-chat as the backbone.

| Method | MedQA (English) | MedQA (Chinese) | IgakuQA (Russian) | RuMedDaNet (Spanish) | MedMCQA (French) | HeadQA (Japanese) |
|---|---|---|---|---|---|---|
| KG-SFT$_{w/o\ Extractor}$ | 37.65 | 39.01 | 40.33 | 65.23 | 25.08 | 23.63 |
| KG-SFT$_{w/o\ Generator}$ | 36.22 | 38.02 | 41.61 | 66.40 | 23.79 | 27.13 |
| KG-SFT$_{w/o\ Detector}$ | 37.24 | 40.05 | 41.61 | 67.66 | 26.52 | 25.13 |
| KG-SFT | **39.31** | **41.71** | **44.40** | **68.75** | **28.45** | **28.14** |

SFT$_{w/o\ Extractor}$, KG-SFT without *Generator*, i.e., without LLMs, KG-SFT directly utilizes triples without converting them into natural language form, as KG-SFT$_{w/o\ Generator}$, and KG-SFT without *Detector*, i.e., without *Detector* to alleviate knowledge conflict, as KG-SFT$_{w/o\ Detector}$, respectively.

As shown in Table 4, the absence of any component within KG-SFT results in a performance degradation of the entire framework. Notably, the absence of *Extractor* has a more significant impact on the performance of KG-SFT, which demonstrates the importance of extracting reasoning subgraphs via external knowledge graphs to promote better comprehension during the SFT process.

## 5.6 RESULTS ON MULTI DOMAINS

Table 5: Accuracy results of SFT, GPT3Mix, AugGPT, TOG, KGR, KAPING and KG-SFT across multi-domains. For each domain, we **bold** the best results and underline the suboptimal ones.

| Domain | SFT | GPT3Mix | AugGPT | TOG | KGR | KAPING | StructGPT | KG-SFT |
|---|---|---|---|---|---|---|---|---|
| Nutrition | 51.29 | 56.45 | 59.68 | 43.55 | 45.16 | 40.32 | 50.84 | **62.35** |
| Astronomy | 48.39 | 49.39 | 50.01 | 38.71 | 41.94 | 35.48 | 49.50 | **54.84** |
| Microeconomics | 39.58 | 41.67 | 41.67 | 35.42 | 39.58 | 29.17 | 34.15 | **47.92** |
| Formal Logic | 38.46 | 42.31 | 34.62 | 31.12 | 32.65 | 34.42 | **45.63** | 39.61 |
| Computer Security | 55.00 | 40.00 | 55.00 | 45.00 | 45.00 | 45.00 | 50.00 | **60.00** |
| Psychology | 45.53 | 47.97 | 44.72 | 45.08 | 44.26 | 43.09 | 45.32 | **52.03** |
| Professional Accounting | 49.12 | 47.36 | **50.87** | 42.55 | 46.18 | 47.24 | 48.88 | 49.62 |
| International Law | 74.00 | 82.00 | 84.00 | 72.00 | 72.00 | 68.00 | 65.00 | **88.00** |
| Management | 70.00 | 71.43 | 72.67 | 57.14 | 52.38 | 57.14 | 68.42 | **75.00** |
| History | 62.50 | 52.08 | 58.33 | 52.64 | 51.32 | 49.50 | 47.18 | **67.08** |
| Professional Law | 40.07 | 43.00 | 39.41 | 34.31 | 34.43 | 32.35 | 36.45 | **43.65** |
| Commensense Reasoning | 55.50 | 63.00 | 62.90 | 59.20 | 57.20 | 61.20 | 62.16 | **64.50** |
| **Avg** | 52.45 | 53.06 | 54.49 | 46.39 | 46.84 | 45.25 | 50.29 | **58.72** |

To demonstrate the generalizability of our approach, we broaden the scope of our datasets. Specifically, we have incorporated over 10 diverse domain datasets from the multi-task language understanding benchmark (Hendrycks et al., 2020). As shown in Table 5, the experimental results indicate that our KG-SFT consistently achieves state-of-the-art performance across most domains, when compared to other data augmentation and knowledge-enhanced methods. In addition, our method has achieved suboptimal results in formal logic and professional accounting. These domians require precise numerical computation or symbolic reasoning, such as mathematics or logical reasoning, where the emphasis is less on knowledge-based inference. This is one direction for the future.

## 5.7 FACTUAL KNOWLEDGE RECALL

To explore the capacity of LLM in manipulating (e.g., recall) factual knowledge within models, we conduct an interpretability analysis following (Yu et al., 2023). This work primarily investigated methods for locating factual knowledge in LLMs, particularly mapping the key neurons (also known as Important Subvalues) to the vocabulary space.

We conduct case analyses and select the representative case as Table 6. More cases can be found in Appendix B.3. For questions related to "cold", the KG-SFT model could directly recall related terms such as "cold", "Cold", "flu", and related to illness states like "sick", "ill", "Ill", and even cause-related terms such as "vir", "virus". In contrast, the knowledge recalled by the original SFT model is mostly unrelated to "cold", and even included special characters like "➡". Overall, KG-SFT performs well on multiple datasets, probably because it provides a lot of correlation and logic of knowledge, which enhances the LLM's ability to recall and locate relevant knowledge during pre-training.

Table 6: Important subvalues' top10 tokens on vocabulary space. Please note that for each model, we analyze the top2 neurons (also known as Important Subvalues) that have the greatest impact on answering the question. In the table, $\text{ffn}_{19}^{2683}$ represents the 2683th neuron located at the 19th MLP layer.

| | Input Text | Probing Token |
|---|---|---|
| | A common viral respiratory infection presenting symptoms like sneezing, sore throat, and runny nose is | cold |

| Method—subvalue | Top Tokens |
|---|---|
| SFT—$\text{ffn}_{31}^{6404}$ | partially, designated, swing, phys, direct, regularly, straight, controlled |
| SFT—$\text{ffn}_{19}^{2683}$ | ➡ , eign, lak, Alo, haupt, ufen, eclipse, isie, illing, hmen |
| KG-SFT—$\text{ffn}_{31}^{6404}$ | partially, designated, phys, swing, direct, regularly, straight, potentially |
| KG-SFT—$\text{ffn}_{21}^{4355}$ | cold, Cold, sick, ill, vir, col, Ill, flu, resp, virus |

## 5.8 COMMONSENSE MULTI-HOP REASONING

Table 7: Commonsense Multi-Hop Reasoning on 3-hop Meta QA

| 3-hop Meta QA | Semantic Similarity | Accuracy |
|---|---|---|
| GPT-3.5 | 66.10 | 53.0 |
| GPT-4o | 67.49 | 55.0 |
| OpenAI o1 | 34.28 | 58.0 |
| SFT | 80.25 | 55.5 |
| AugGPT | 83.00 | 62.9 |
| GPT3Mix | 83.35 | 63.0 |
| Think on Graph | 81.20 | 59.2 |
| KGR | 80.79 | 57.2 |
| KAPING | 81.79 | 61.2 |
| KG-SFT | **84.25** | **64.5** |

To explore the LLMs' knowledge reasoning ability and demonstrate that the remarkable performance of KG-SFT is not limited to specific domains, we conduct experiments on the common sense question-answering dataset Meta QA (Zhang et al., 2018). Specifically, Meta QA is a multi-hop reasoning question-answering dataset. We select the most complex 3-hop questions from it as our experimental data. As shown in Table 7, KG-SFT significantly outperforms the baselines in both semantic similarity and accuracy of the answers, and even the strong baseline OpenAI o1. For example, the accuracy of KG-SFT achieves a notable increase of 9.0% compared to SFT. In summary, KG-SFT continues to achieve remarkable results in the domain of common sense and can enhance the LLMs' multi-hop reasoning capabilities. This may be one of the reasons behind the superior performance of KG-SFT.

## 6 CONCLUSION AND DISCUSSION

In this paper, we propose a conceptually flexible, and general framework **K**nowledge **G**raph-Driven **S**upervised **F**ine-**T**uning that focuses on **quality** augmentation to boost supervised fine-tuning. Specifically, we propose *extractor*, *generator*, and *detector* to generate high-quality explanations for each Q&A pair via structured knowledge graph to promote better **knowledge manipulation** for LLMs. Extensive experiments demonstrate the effectiveness of our KG-SFT, leading to a maximum accuracy improvement of up to 18.1% and an average of 8.7% in low-data scenarios. Moreover, KG-SFT also serves as a plug-and-play framework for existing **quantity** augmenting methods that achieve a maximum relative improvement of 88.71% in the accuracy metric and achieve the new state-of-the-art methods.

## 7 ETHICS STATEMENT

This paper presents the Knowledge Graph-Driven Supervised Fine-Tuning (KG-SFT) framework to enhance large language models (LLMs) in specific domains. Our research adheres to ethical guidelines, avoiding human subjects or sensitive data. The data used consists solely of open source SFT data, with no harmful applications identified. While KG-SFT aims to improve knowledge comprehension and manipulation, we discourage the use of the generated models in high-stakes scenarios without further validation, as the potential for errors or misinterpretations exists. No conflicts of interest were found, and all experiments comply with relevant ethical standards.

## 8 REPRODUCIBILITY STATEMENT

In this study, to ensure the reproducibility of our approach, we provide key information from the main text and Appendix as follows.

1. **Algorithm and Experimental Details.** We provide the architecture of our approach **KG-SFT** in Section 4. We also provide the detailed implementation of **KG-SFT** in Appendix A. See Appendix A.4 for the PROMPTS of KG-SFT. Moreover, we provide detailed experiment settings in Section 5.1, Appendices A.1, A.2, and A.3.

2. **Source Code.** According to the architecture in Section 4, the BM25 algorithm, HIST algorithm, NER tools, and training framework we used are all open-source and publicly available. Specifically, in Section 5.1, we use the code from (Zheng et al., 2024) for model training, available at https://github.com/hiyouga/LLaMA-Factory. Moreover, we are committed to providing the source code of our approach, if accepted.

ACKNOWLEDGMENTS

The authors would like to thank all the anonymous reviewers for their valuable suggestions. This work was supported by the National Key R&D Program of China under contract 2022ZD0119801 and the National Nature Science Foundations of China grants U23A20388 and 62021001.

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

# A  IMPLEMENTATION

In this section, we introduce the implementation details of the experiments, including training parameters and prompts used.

## A.1  DATASET DETAILS

Table 8 presents the statistical results for medical multiple-choice questions benchmarks in six language.

Table 8: Statistical results for medical multiple-choice questions benchmarks in six languages.

| Dataset | Language | Source | Train | Test |
|---------|----------|--------|-------|------|
| MedQA | English | United States Medical Licensing Examination | 10178 | 1273 |
| MedQA | Chinese | United States Medical Licensing Examination | 27400 | 3426 |
| IgakuQA | Japanese | Japan's medical licensure exams (2018-2022) | 1590 | 199 |
| RuMedDaNet | Russian | Russian medical judgment question dataset | 1052 | 256 |
| FrenchMedMCQA | French | Professional exams for the French Pharmacy degree | 2171 | 622 |
| Head-QA | Spanish | Exams for positions in the Spanish healthcare | 2657 | 2742 |

## A.2  TRAINING DETAILS

Specifically, we use two data formats, the vanilla SFT data without explanations and the enhanced KG-SFT data with explanations, to conduct full-model fine-tuning training. In the fine-tuning phase, our optimization objective is minimizing the loss between generated text and target text. We set the maximum context length to 2048, padding each batch to match the longest sequence in that batch. We use AdamW optimizer with the following hyper-parameters: $\beta_1 = 0.95, \beta_2 = 0.9$. For full-model fine-tuning, we utilized DeepSpeed, BF16 data type, and gradient checkpointing technology. We set the global batch size to 64 and the warmup ratio to 0.03. For vanilla SFT data without explanations, we set a learning rate of 1e-6. In the case of the enhanced KG-SFT data with explanations, we set a learning rate of 5e-6. Finally, the models are trained on four A100 GPUs for 5 epochs.

## A.3  FINE-TUNING PROMPTS

It is worth noting that the two SFT data formats contain different types of Q&A data. The vanilla SFT data without explanations only contains instructions that only require the correct answer for each Q&A pair. In the KG-SFT data, for each Q&A pair, there are not only instructions that only require the correct answer, but also instructions that require the model to give the explanations.

In our fine-tuning approach, we employ two distinct types of prompts for the two instructions. This helps the models discern whether they should generate detailed rationale sentences or not, thus minimizing confusion when the inference phase only requires the model to give the correct answer. Specifically, for the instructions that only require the correct answer, we use the following prompt:

```
Please play the role of a language doctor, respond to the
medical inquiries based on the patient's account.  Provide
the most appropriate option directly.
```

In contrast, to obtain an answer accompanied by its corresponding rationale, we use a more detailed prompt:

```
Please play the role of a language doctor, respond to
the medical inquiries based on the patient's account.
Provide the most appropriate option directly.  Let's solve
this step-by-step.  You should first give the reason in
{language} for your choice.  Then you should give the right
answer.
```

It's important to note that during evaluation phase, we only need the model to give the correct answer to calculate the accuracy for the multiple-choice questions. So we used the first instruction prompt for the inference.

## A.4 KG-SFT Prompts

In the generator and detector components of KG-SFT, we utilize LLMs to accomplish the specified tasks. The details of the prompts used are illustrated in the table9.

Table 9: The prompts used in KG-SFT. In the prompts, "{str(qa)}" represents a specific Q&A pair and "{str(triples)}" represents the reasoning subgraph obtained from extractor.

| Prompt Type | Text |
| --- | --- |
| Prompts for generating explanations in the generator | Assuming you are a knowledgeable and experienced medical expert, please generates a logical and fluent explanation based on the knowledge graph information (triple list) provided below, as well as the questions and answers, and be careful not to mention "knowledge graphs" or "triple" in the output, as these contents are only visible to you.
Question and answer:{str(qa)}
Triples for reasoning subgraph:{str(triples)}
The generated format is json like this: {"Explanation": "..."}.
You should output with {language} and do not output any irrelevant content. |
| Prompts for re-generating in the detector | Assuming you are a knowledgeable and experienced medical expert, the explanation below contains content that conflicts with the knowledge graph (sentences with knowledge conflicts have been marked with an * on both sides). Q&A, explanation, and related knowledge graphs are as follows. Please generate the correct explanation again, and be careful not to mention "knowledge graphs" or "triple" in the output, as these contents are only visible to you.
Question and answer:{str(qa)}
Triples for reasoning subgraph:{str(triples)}
Explanation:{str(explanation)}
The generated format is josn like this: {"Explanation": "..."}.
You should output with {language} and do not output any irrelevant content. |

## B More Results

### B.1 Investigate the potential for data leakage

To further investigate the potential for data leakage, we analyze whether the performance improvements of the model on the test set are directly attributable to the KG content added to the training set. We conduct a thorough analysis to ensure that our experiments are not affected by such potential issues. Firstly, we employed a state-of-the-art Natural Language Inference (NLI) model, DeBERTa, to assess the semantic relationship between each generated explanation and every question in the test set. Specifically, we categorized the relationships as entailment, neutral, or contradiction.

(i) An **entailment** indicates that the generated explanation directly answers the test question.

(ii) A **neutral** indicates no direct semantic connection.

(iii) A **contradiction** indicates a semantic conflict.

Our results revealed that only 0.01% of the explanations were classified as entailment, while 97.71% were classified as neutral, and 2.28% as contradiction. This suggests that our performance improvements are not attributable to data leakage, and the presence of contradictions aligns with the claim in our paper that knowledge conflicts may still occur in generated explanations.

Table 10: The **data leakage ratio** for different values of overlap $k$. If the proportion of overlap between the entity set of a question in the test set and the entity set of any question in the training set exceeds $k$, that data point is considered to have potential data leakage.

| Overlap-k | 0.2 | 0.25 | 0.3 | 0.35 | 0.4 |
|---|---|---|---|---|---|
| **data leakage (%)** | 1.18 | 0.31 | 0.16 | 0.00 | 0.00 |

Furthermore, we conducted a statistical analysis to check for potential overlap between the entities in the training and test sets. We performed Named Entity Recognition (NER) on each question in both sets. We defined a threshold K to evaluate if there was significant overlap between entities in test set questions and any training set questions. As shown in Table 10, with K set at 0.2, we found that only 1.18% of the test set questions showed potential overlap with the training set. Increasing K to 0.35 resulted in no detectable overlap. **These results provide additional statistical evidence supporting that there is no significant data leakage between our training and test sets**.

### B.2 RESULTS OF COMPUTATIONAL OVERHEAD

Table 11: Time comparison of SFT, GPT3Mix, and KG-SFT at different augmentation ratios.

| Ratio | Num. | Time (min) | | |
|---|---|---|---|---|
| | | **SFT** | **GPT3Mix** | **KG-SFT** |
| 5% | 506 | 3 | 7 | 5 |
| 20% | 2032 | 12 | 25 | 18 |
| 50% | 5081 | 25 | 60 | 40 |
| 100% | 10128 | 52 | 120 | 85 |

We conducted experiments to evaluate the computational overhead of our KG-SFT method compared to vanilla supervised fine-tuning in Table 11. Our experiments were performed using 4 A100 GPUs(80GB) over 5 epochs with the LLaMA2-7B model. Notably, our approach only involves data synthesis, and thus, the overhead is independent of model size. According to our results, KG-SFT incurs approximately 1.5 times the computational overhead of the original SFT, while typical data augmentation methods, such as GPT3Mix (which doubles the dataset size), result in an overhead of around 2 times.

### B.3 MORE CASES FOR FACTUAL KNOWLEDGE RECALL

In Table 12, we provide more cases for experiments of factual knowledge recall.

### B.4 MULTILINGUAL TRANSFER EXPERIMENTS

To further explore whether KG-SFT can enhance the knowledge transfer capability of LLMs, we conduct multilingual transfer experiments. Specifically, as shown in Figures 2 and 3, the y-axis represents the language type of the fine-tuning data, and the x-axis represents the language type of the test data. This setup is used to investigate if the knowledge or abilities acquired through fine-tuning in one language can be transferred to another language. For a clearer comparison, we list the performance comparison between KG-SFT and SFT in Table 13, where each value represents the accuracy difference between KG-SFT and SFT. From able 13, it is demonstrated that KG-SFT

Table 12: Important subvalues' top10 tokens on vocabulary space. Please note that for each model, we analyze the top2 MLPs (also known as Important Subvalues) that have the greatest impact on answering the question. In the table, $\text{ffn}_{18}^{1105}$ represents the 1105th MLP located at the 18th layer.

| Input Text | Probing Token |
|---|---|
| The disease characterized by the growth of abnormal cells in the lungs is | cancer |

| Method—subvalue | Top Tokens |
|---|---|
| SFT—$\text{ffn}_{18}^{1105}$ | prost, suic, sexual, murder, sex, drug, dru, cancer, Blood, assass |
| SFT—$\text{ffn}_{10}^{0802}$ | squ, sar, mel, cancer, car, mes, colon, tum, onc, rare |
| KG-SFT—$\text{ffn}_{18}^{1105}$ | prost, suic, sexual, sex, murder, drug, dru, cancer, Blood, lung |
| KG-SFT—$\text{ffn}_{16}^{0801}$ | clin, surg, patients, disease, patient, medic, medicine, drug, cancer, medical |

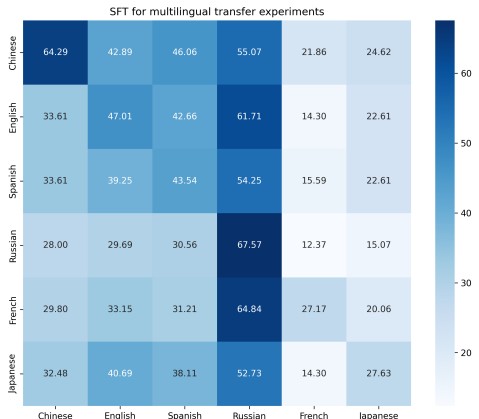

Figure 2: SFT for Multilingual Transfer Experiments

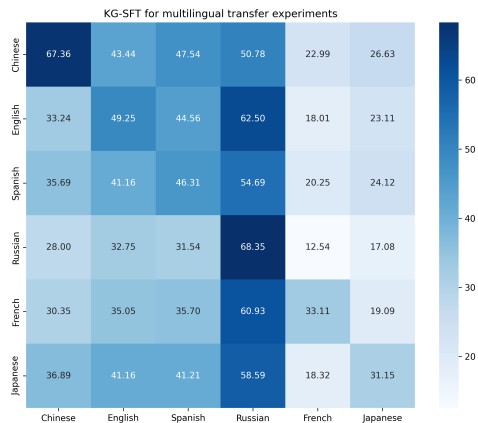

Figure 3: KG-SFT for Multilingual Transfer Experiments

Table 13: Performance comparison between KG-SFT and SFT

|  | Chinese | English | Spanish | Russian | French | Japanese |
|---|---|---|---|---|---|---|
| Chinese | +4.77% | +1.28% | +3.21% | -7.79% | +5.17% | +8.20% |
| English | -1.10% | +4.82% | +4.45% | +1.28% | +25.87% | +2.21% |
| Spanish | +6.21% | +4.86% | +6.35% | +0.81% | +29.90% | +6.68% |
| Russian | +0.00% | +10.31% | +3.20% | +1.15% | +1.38% | +13.34% |
| French | +1.85% | +5.73% | +14.38% | -6.03% | +21.90% | -7.33% |
| Japanese | +13.58% | +1.16% | +8.13% | +11.12% | +28.04% | +12.72% |

outperforms SFT in the majority of indicators. For example, in the transfer from Japanese to Russian, KG-SFT achieves an accuracy rate of 58.59, compared to SFT's 52.73, marking an actual improvement of 5.86%, or a relative improvement of 11.12%. Moreover, in the transfer to Russian and Japanese, KG-SFT does not consistently outperform SFT. Referring to the (Touvron et al., 2023b), we discover that in the pre-training corpus of Llama 2, Russian made up only 0.13% and Japanese a mere 0.10%, significantly less than other languages. This suggests that the likely reason is that Llama 2 stored less relevant knowledge during pre-training for these languages. In summary, KG-SFT demonstrates a superior transfer capability compared to the original SFT, which might be one of the reasons for its better performance.

## B.5 RESULTS OF DIFFERENT LLMS

We also conduct experiments to demonstrate the generalizability of various LLMs. We apply general LLMs (LLaMA-2-7B-chat and BLOOMZ-7B-chat) and medical LLMs (MMedLM2) as the backbone

Table 14: Experiment results for different LLM backbones, including LLaMA-2-7B-chat, BLOOMZ-7B-chat, and MMedLM2 7B.

| Model | Metric | Chinese | English | Spanish | Russian | French | Japanese |
|---|---|---|---|---|---|---|---|
| Llama 2 | sft | 33.62 | 29.33 | 66.40 | 35.19 | 12.67 | 21.11 |
| | kgsft | 41.71 | 39.31 | 68.75 | 44.40 | 28.45 | 28.14 |
| | Impr. | +24.06% | +34.02% | +3.54% | +26.15% | +124.54% | +33.29% |
| BLOOMZ | sft | 41.09 | 32.60 | 37.61 | 58.59 | 12.86 | 19.59 |
| | kgsft | 43.72 | 36.99 | 41.24 | 60.15 | 21.22 | 25.12 |
| | Impr. | +6.41% | +13.45% | +9.66% | +2.66% | +65.00% | +28.23% |
| MMedLM2 | sft | 63.45 | 50.82 | 59.4 | 67.18 | 28.29 | 46.73 |
| | kgsft | 69.61 | 57.34 | 64.29 | 78.12 | 48.55 | 58.29 |
| | Impr. | +9.71% | +12.81% | +8.23% | +16.28% | +71.65% | +24.75% |

models. As shown in Table 14, we can observe that KG-SFT significantly outperforms the traditional SFT method across all language settings. Specifically, in the French scenario, KG-SFT gets relative improvement by 124.54% compared to the vanilla SFT method. For MMedLM2, our KG-SFT still maintains consistent performance improvements across all languages. These results further demonstrate the generalizability of KG-SFT over various LLM backbones, which highlights the importance of generating explanations for Q&A pairs.

## B.6 MORE RELATED WORK

Supervised fine tuning (SFT) is a powerful alignment technique for LLMs, which can help LLMs adapt to specialized domain tasks or align with human intentions. SFT can also refer to general sequence-to-sequence fine-tuning, which includes human alignment, instruction fine-tuning, downstream task fine-tuning, etc (Dong et al., 2023). Recent research explores multi-task SFT to achieve better zero-shot performance across various downstream tasks (Sanh et al., 2021). (Chung et al., 2024) and (Longpre et al., 2023) further integrate almost all existing NLP tasks for large-scale multi-task instruction fine-tuning. Moreover, some methods attempt to apply SFT to more complex downstream tasks such as mathematical reasoning (Yuan et al., 2023; Hendrycks et al., 2021; Liu et al., 2024a) and code generation (Luo et al., 2023). Regarding research on knowledge graphs, some early work focused on reasoning within knowledge graphs to enhance link prediction (Sun et al., 2019; Zhang et al., 2020) and inductive link prediction (Teru et al., 2020; Chen et al., 2021; Liu et al., 2024b) capabilities. Although these efforts do not directly enhance the reasoning abilities of LLMs, they can provide potential insights for future directions in augmenting LLMs with knowledge graphs. Moreover, another line integrates pre-trained LLM and graph neural networks (Shi et al., 2023; 2025), to encode the texts and graph structures simultaneously (Shi et al., 2024; Li et al., 2024b).

## B.7 MORE RESULTS OF LLM SCORER METHODS

We further have conducted additional experiments by replacing the HITS scoring algorithm with semantic-based scoring methods using LLaMA 2 models (13B and 70B) for entity selection to provide a more comprehensive insight of our KG-SFT. As shown in Table 15, the LLaMA 2 70B model achieved notable performance, even surpassing the original HITS-based KG-SFT in certain test cases, which highlights the effectiveness of semantic scoring approaches. However, when considering overall accuracy, the HITS algorithm still delivered the best results while also being significantly more cost-efficient. These findings further validate the rationale behind our choice of the HITS algorithm. In future work, we can explore and optimize scoring methods further, particularly in the context of downstream task requirements, to strike a balance between accuracy, interpretability, and computational efficiency.

## B.8 MORE RESULTS FOR 70B

Table 15: More results for LLM scorer variants methods. We **bold** the best results for each dataset.

| Method | #Tuning QA | MedQA (English) | MedQA (Chinese) | IgakuQA (Russian) | RuMedDaNet (Spanish) | MedMCQA (French) | HeadQA (Japanese) | Avg. |
|---|---|---|---|---|---|---|---|---|
| KG-SCORE-13B | 1000 | 41.79 | 37.91 | 68.75 | 43.28 | 25.72 | 27.11 | 40.76 |
| KG-SCORE-70B | 1000 | **42.41** | 38.32 | **71.48** | 43.10 | 25.72 | **29.14** | 41.70 |
| KG-SFT | 1000 | 41.71 | **39.31** | 68.75 | **44.40** | **28.45** | 28.14 | **41.79** |

We further have conducted additional experiments with the LLaMA-2-70B model. As shown in Table 16, the results demonstrate that KG-SFT continues to achieve the best performance and significant improvements even at this larger scale. Moreover, in some test cases (MedQA-C, IgakuQA, and MedMCQA), the KG-SFT-enhanced 7B model even outperforms the original and SFT-finetuned LLaMA-2-70B models. These findings strongly validate the effectiveness and scalability of the KG-SFT approach.

Table 16: Comparison of methods on 70B

| Method | QA_num | English | Chinese | Russian | Spanish | French | Japanese | Avg |
|---|---|---|---|---|---|---|---|---|
| Vanilla-7B | - | 28.20 | 28.37 | 51.17 | 32.97 | 12.76 | 11.10 | 27.43 |
| SFT-7B | 1000 | 33.62 | 29.33 | 66.40 | 35.19 | 12.67 | 21.11 | 33.05 |
| KG-SFT-7B | 1000 | 41.71 | 39.31 | 68.75 | 44.40 | 28.45 | 28.14 | 41.79 |
| Vanilla-70B | - | 45.99 | 34.67 | 50.00 | 47.38 | 20.45 | 20.10 | 36.43 |
| SFT-70B | 1000 | 47.89 | 37.53 | 66.52 | 48.10 | 22.55 | 30.80 | 42.23 |
| **KG-SFT-70B** | **1000** | **51.21** | **46.73** | **69.53** | **48.94** | **30.06** | **36.68** | **47.19** |

## C  CASE STUDY

In this section, we present a detailed analysis of individual cases within the English dataset. Specifically, we compare the responses generated by various models, including the vanilla Llama2 model, the Llama2 model fine-tuned with the vanilla SFT, and the Llama2 model fine-tuned using KG-SFT. Through this comparative analysis, we aim to demonstrate the superior performance of the KG-SFT method. The specific results of this comparison are illustrated in the accompanying figures 4,5,6,7.

In Case 1, all models answer correctly, including the vanilla Llama2 model that did not undergo SFT. This question involves how residents should document surgical reports, specifically emphasizing that all intraoperative events must be accurately recorded. This represents a straightforward assessment of professional medical knowledge with simple logic. In the knowledge graph, this constitutes one-to-one single-hop logical reasoning, which all models can easily handle.

In Case 2, the vanilla Llama2 model did not provide the correct answer, but both the vanilla SFT-trained and KG-SFT-trained models did. This question presents the patient's symptoms and asks for the most likely diagnosis. The symptoms of hyperandrogenism, menstrual irregularities, obesity, and glucose intolerance all indicate PCOS. In the knowledge graph, this represents many-to-one single-hop logical reasoning. Due to the fragmented nature of the knowledge required, the vanilla Llama2 model could not answer correctly. However, since this question still belongs to single-hop reasoning, the vanilla SFT-trained model are able to provide the correct answer.

In Case 3, only the model trained with KG-SFT can provide the correct answer. This question presents the patient's symptoms and asks what additional symptoms the patient may experience. To address this, it is necessary to first diagnose the patient's disease based on the initial symptoms, and then predict other potential symptoms associated with the diagnosed disease. This process involves many-to-one and one-to-many multi-hop reasoning within the knowledge graph. The vanilla SFT model fails to solve this problem, whereas KG-SFT successfully provides the correct answer. This demonstrates that our method enhances the model's capability for multi-hop reasoning and knowledge manipulation.

In Case 4, none of the models answered correctly. The problem analysis revealed that the patient had diabetes and peripheral arterial disease, which might suggest consideration of vascular-related diseases. Additionally, right-sided flank pain and hypertension can be associated with various

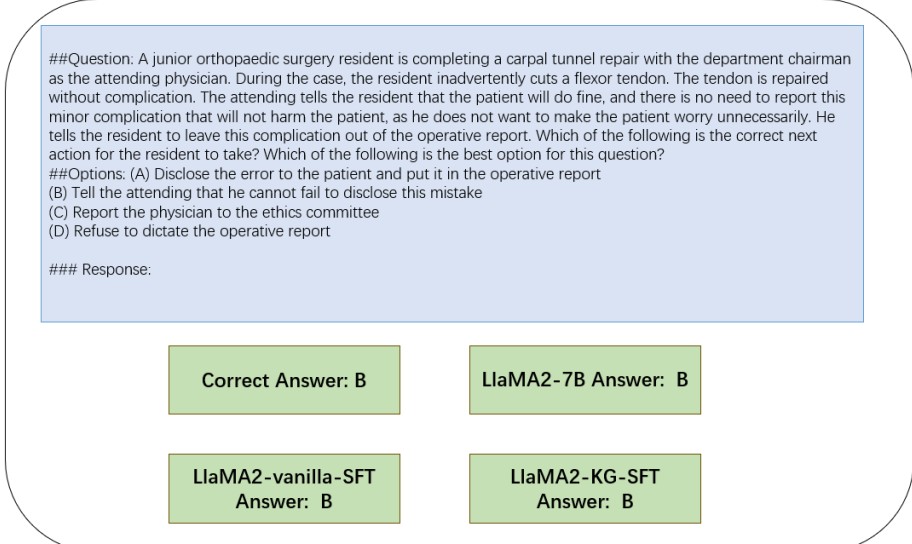

Figure 4: Case 1. In this example, all models can answer correctly, even the vanilla Llama2 model that did not pass SFT.

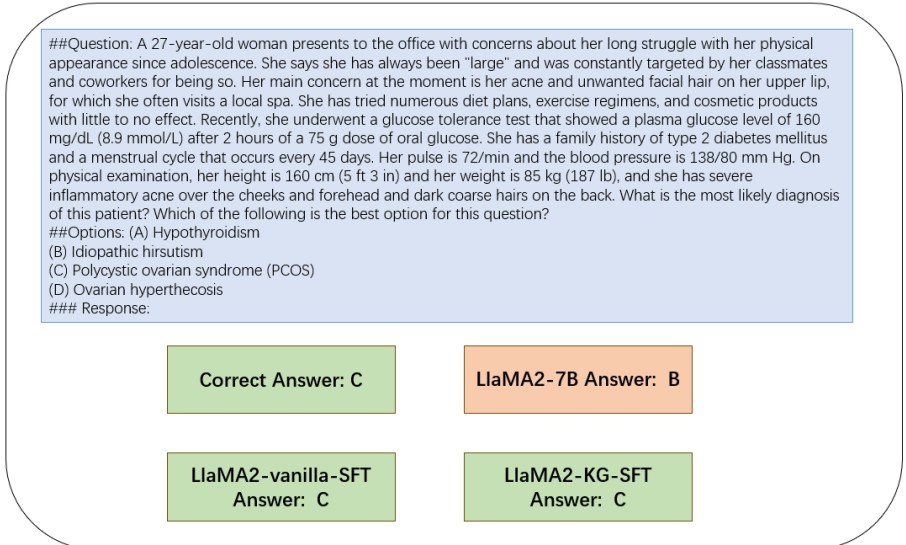

Figure 5: Case 2. In this example, the vanilla llama2 model don't get it right. Both SFT trained and KG-SFT trained models answer correctly.

conditions, complicating the diagnosis. The causes of dilation of the right ureter and renal pelvis (i.e., hydronephrosis) are diverse and necessitate comprehensive judgment based on clinical manifestations. This question requires identifying multiple possible causes and conducting a thorough analysis based on the patient's specific symptoms and examination results. Even in real-life medical scenarios, this question is still a very difficult one. The models, including KG-SFT, still struggle to answer such inductive questions accurately.

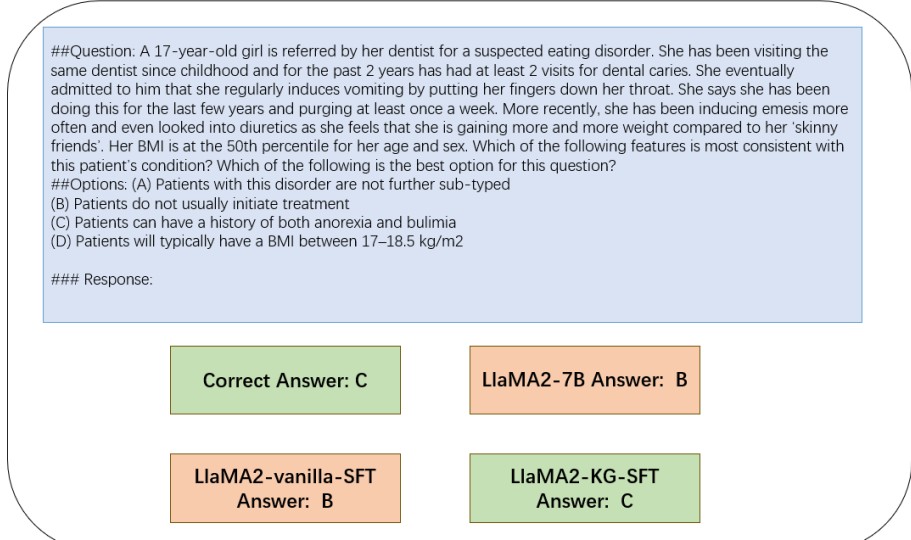

Figure 6: Case 3. In this example, only the model trained by KG-SFT answers correctly

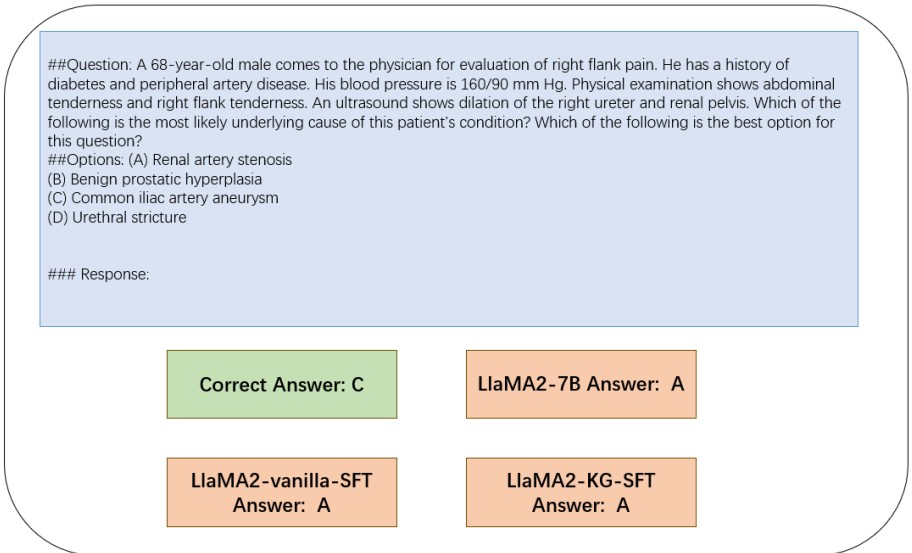

Figure 7: Case 4. In this example, none of the models answer correctly.

