# OpenReview forum: "Knowledge Graph Finetuning Enhances Knowledge Manipulation in Large Language Models"
_ICLR.cc/2025/Conference — ICLR 2025 Poster_

### Official Review · Reviewer_6CzA · 2024-10-29

**Soundness:** 3
**Presentation:** 3
**Contribution:** 2
**Rating:** 6
**Confidence:** 4

**Summary:**

Existing supervised fine-tuning (SFT) data requires the supplementation of fact triples from knowledge graphs. To bridge the gap between the correlations and knowledge underlying the SFT question-answer (QA) pairs, this paper proposes a conceptually flexible and general framework, referred to as knowledge graph-driven supervised fine-tuning (KG-SFT). This architecture comprises three components: Extractor, Generator, and Detector. First, the Extractor derives relevant reasoning subgraphs from the knowledge graph based on the QA pair to uncover the underlying correlation and logic of the knowledge. Next, the Generator uses a large language model (LLM) to create explanations for the given QA pair and transform the structured knowledge and logic into a natural language format. Finally, the Detector examines these explanations against the triples from the inference graph to ensure accuracy. To verify the framework's effectiveness, the authors conduct extensive experiments, including testing across datasets in six languages and 15 domains.

**Strengths:**

1 The authors present a general framework that incorporates fact triples from knowledge graphs to enhance LLMs' ability to handle knowledge-intensive tasks, such as knowledge base question answering.

2 The authors conduct various experiments to support their claims, including comparisons between KG-SFT and SFT, assessments of KG-SFT's superiority and generalizability, and evaluations of the LLM's reasoning and knowledge transfer capabilities under the proposed framework. These experiments are comprehensive and praiseworthy.

**Weaknesses:**

1 The architecture is complex. Specifically, the Extractor uses a named entity recognition (NER) model, such as Metamap, to extract entity lists from questions and an off-the-shelf BM25 model to rank triples. The Generator applies an off-the-shelf HITS model to filter significant content within the reasoning subgraph. The Detector uses an off-the-shelf natural language inference (NLI) model to identify knowledge conflicts. Throughout this process, additional information filtering (e.g., selecting the top 20 relevant triples in the Extractor) is employed, which adds considerable complexity to the overall architecture.

2 Despite its complexity, the authors do not provide their code or a detailed README file, which may hinder reproducibility and ease of use.

**Questions:**

1 In the Extractor component, a NER model is used to obtain the entity list from the questions. How does the system handle cases where recognized entities are absent in the knowledge graph?

2 I assume this architecture is used for data augmentation, where the processed data trains the LLM, which is then fine-tuned to answer questions. Is this accurate? Or, in the Detector component, does the LLM directly answer questions without additional fine-tuning?

3 In Section 5.7, what method is used to identify the top two neurons?

4 In Section 5.8, experiments are conducted on the MetaQA dataset, a knowledge graph QA dataset. The case study in the Appendix suggests that LLMs are not provided with the triples from the knowledge graph, whereas your method retrieves facts from the graph. Consequently, I wonder if the experiments in Table 7 are fair, as GPT and o1 should also have access to the retrieved triples.

5 Entities within MetaQA questions are annotated, which implies that the Extractor component might not play a role in this experiment. Could you clarify?

---

### Official Review · Reviewer_HEoP · 2024-10-31

**Soundness:** 3
**Presentation:** 2
**Contribution:** 3
**Rating:** 6
**Confidence:** 4

**Summary:**

This paper proposes a conceptually flexible and universal framework to enhance SFT, namely Knowledge Graph-driven Supervised Fine-Tuning (KG-SFT). The core idea of KG-SFT is to generate high-quality explanations for each Q&A pair through a structured knowledge graph, thereby enhancing LLMs' understanding and manipulation of knowledge. KG-SFT consists of three components: Extractor, Generator, and Detector. For a given Q&A pair, (i) the Extractor first identifies entities within the Q&A pair and extracts relevant reasoning subgraphs from an external knowledge graph; (ii) the Generator then uses these reasoning subgraphs to generate corresponding fluent explanations; (iii) finally, the Detector performs sentence-level knowledge conflict detection on these explanations to ensure their reliability. The authors conducted extensive experiments to demonstrate the effectiveness of KG-SFT.

**Strengths:**

Originality : The author has proposed a flexible and universal framework, KG-SFT, to enhance SFT. This framework generates high-quality explanations for each Q&A pair through a structured knowledge graph, aiming to improve the understanding and manipulation capabilities of large language models (LLMs). The KG-SFT framework includes a three-stage process design consisting of the Extractor, Generator, and Detector.

Quality : The experiments are extensive and ample. Tests were conducted on datasets in six languages, compared with 12 baselines, and joint experiments on quantity and quality enhancement were carried out. Ablation studies were performed on each component, incorporating datasets from more than 10 different fields from the multi-task language understanding benchmark as well as datasets from over 10 different domains.

Clarity : The author provides a clear exposition of the innovations, methods, and experiments.

Significance : This paper may provide some references for existing knowledge-enhanced fine-tuning methods and promote the development of the field.

**Weaknesses:**

1.	Some modules in the process are designed to be quite basic. During the Extractor phase, open-source tools are used for Named Entity Recognition (NER). In the Generator phase, Large Language Models (LLMs) are utilized to transform the structured knowledge and logic behind the questions into natural language text format. These steps appear to be quite routine and lack innovation.

2.	The use of the HITS algorithm for scoring entities during the Generator phase may seem somewhat outdated. The weight scores obtained from the HITS algorithm do not always accurately reflect the importance of entity nodes. Considering this, it may be worthwhile to explore methods that focus more on semantics or have greater interpretability for scoring entities, such as using Large Language Models (LLM) for scoring. Has the author considered conducting comparative experiments with different scoring methods to highlight the effectiveness of each approach?

3.	MetaQA is a relatively simple dataset where many methods can achieve performance above 90% on 3-hop questions, such as KG-GPT at 94.0%, KB-BINDER at 96.4%, and UniKGQA at 99.1%. However, the methods compared in Section 5.8, as well as KG-SFT, seem to lag significantly behind state-of-the-art models. Could an explanation be provided for this discrepancy?

**Questions:**

See weakness.

---

### Official Review · Reviewer_aU7W · 2024-11-04

**Soundness:** 2
**Presentation:** 3
**Contribution:** 2
**Rating:** 6
**Confidence:** 4

**Summary:**

This paper addresses the challenges large language models face in specific domains, particularly in low-data and knowledge-intensive tasks. It proposes Knowledge Graph-Driven Supervised Fine-Tuning (KG-SFT), a framework that boosts supervised fine-tuning by generating high-quality explanations for Q&A pairs using structured knowledge graphs. KG-SFT comprises three components: Extractor, which identifies entities and extracts reasoning subgraphs; Generator, which creates fluent explanations; and Detector, which ensures reliability by detecting knowledge conflicts. Extensive experiments show the effectiveness of KG-SFT.

**Strengths:**

This paper is well written and easy to understand.
This papaer proposes KG-SFT, which is composed of three components, Extractor, Generator, and Detector. The designed Detector can enhance the correctness of the generated explanations by Generator, which is novel.
This paper conducts extensive experiments to show the improvement of KG-SFT and offer high-quality analytical insights.

**Weaknesses:**

Table 4 shows that all three components, Extractor, Generator, and Detector, contribute to the performance of the model. However, it seems that Extractor and Generator is not new. It is better to explain the innovation of Extractor and Generator clearly.
The paper should analyze the causes of knowledge conflicts. Is the knowledge conflict due to the insufficient ability of the large language model or the inaccurate extraction of the subgraph or the incorrect selction in Generator?
It would be better if the paper could briefly describe the prompts for Generator and Detector in the main paper.

**Questions:**

The second step of Extractor ranks the triples and retains the top related triples. Is it also necessary to do the same in the third step?
In Line 312, why do not Generator and Detector use the same backbone?

---

### Official Review · Reviewer_5WAX · 2024-11-04

**Soundness:** 3
**Presentation:** 3
**Contribution:** 3
**Rating:** 6
**Confidence:** 3

**Summary:**

This paper explores the challenges of LLMs in specific domains, such as low-data and knowledge-intensive Question and Answer (Q&A) tasks. The authors first emphasize the importance of the correlation and logic of knowledge underlying the Q&A and propose the Knowledge Graph-Driven Supervised Fine-Tuning (KG-SFT) framework to construct detailed explanations for Q&A pairs. Experimental results on fifteen different domains and six different languages show that the presented methods are effective.

**Strengths:**

1.	This paper reveals the challenge of existing SFT data in low-data and knowledge-intensive Q&A tasks, which can attract research attention.
2.	This paper proposes the KG-SFT framework, which is effective to construct high-quality data and alleviate the above problems.
3.	The authors conduct the experiments from different perspectives and present the significant performance improvement compared with baselines.

**Weaknesses:**

1. The investigated model sizes are limited to 7B (LLaMA-2-7B-chat, BLOOMZ-7B-chat, and MMedLM2-7B). Larger models, such as 70B models, are worth exploring because they contain broader knowledge. Can the KG-SFT method still improve compared with the SFT approach?
2. In Section 5.8, the details about the prompts for closed-sourced LLMs (eg. GPT-4o, OpenAI o1) should be given. Besides, for multi-hop reasoning tasks, the other prompt-based baselines enhanced with structure data, such as StructGPT[1], could be added.

[1] Jinhao Jiang, Kun Zhou, Zican Dong , Keming Ye, Wayne Xin Zhao and Ji-Rong Wen. StructGPT: A General Framework for Large Language Model to Reason over Structured Data. EMNLP 2023.

**Questions:**

1. The KG-SFT approach seems to rely on the knowledge graph to construct high-quality explanations. Is it suitable for commonsense reasoning tasks, such as CQA[1]?

[1] Alon Talmor, Jonathan Herzig, Nicholas Lourie, and Jonathan Berant. CommonsenseQA: A question answering challenge targeting commonsense knowledge. ACL 2019.

---

### Official Review · Reviewer_o9iR · 2024-11-05

**Soundness:** 2
**Presentation:** 2
**Contribution:** 3
**Rating:** 5
**Confidence:** 3

**Summary:**

This paper aims to enhance knowledge comprehension when fine-tuning the large language models in specific domains (e.g., low-data and knowledge-intensive). To this end, they propose a method (KG-SFT) to leverage the external knowledge graphs to generate high-quality explanations for each training Q-A pair. Starting from the original Q-A pairs, KG-SFT sequentially utilizes three modules (i.e., Extractor, Generator, and Detector) to identify entities, generate explanations, and detect knowledge conflicts.

**Strengths:**

1. The raised problem is interesting and looks convincing to me. LLM fine-tuning usually confronts the knowledge-aware requirements for some specific domains.
2.   Authors conduct enough experiments to verify the proposed methods. The authors also provide a detailed analysis of each experiment.

**Weaknesses:**

1. The idea is very trivial and straightforward, and the innovation of the proposed method is limited. The main motivation of KG-SFT is to add detailed explanations to original Q&A pairs using external KGs. However, there are usually not enough precise KGs with broad coverage in specific domains. This makes the method hard to apply in practice. What's more, the three components mainly leverage traditional rule-based or term-matching-based methods to extract knowledge explanations, which is limited to the correctness of these methods. There is no reasonable explanation about how they guarantee the correctness of external knowledge.
2. The paper written is a little unclear with some missing parts. In section 4, the authors mainly introduce how they generate explanations from KGs, but there is a lack of content about how they fine-tune the LLMs.
3. The selected baselines are simple, and there is no recent SFT-enhancing baseline for comparison.

**Questions:**

The format of some citations is incorrect. For example,  in Line 39 and Line 69, it should be \citep rather than \citet

---

### Meta-Review · Area_Chair_TrDC · 2024-12-20

**Metareview:**

This paper presents a novel framework called Knowledge Graph-Driven Supervised Fine-Tuning (KG-SFT) to address the challenges of supervised fine-tuning in domain-specific applications of LLMs, particularly in low-data and knowledge-intensive settings. KG-SFT enhances the comprehension and manipulation of knowledge in LLMs by generating high-quality explanations for each Q&A pair using structured knowledge graphs. The experiments across fifteen different domains and six languages demonstrate an improvement in performance, validating the effectiveness of KG-SFT.

The reviewers raised several questions about the practicality of applying KG-SFT in domains where precise and broad-coverage knowledge graphs are scarce, the clarity and completeness of the methodology description, particularly the fine-tuning process, and the choice of baselines, which do not include recent SFT-enhancing methods. Concerns were also expressed about the novelty of the individual components of KG-SFT and the validation of its necessity over single-capability setups. During the rebuttal, the authors addressed these concerns by providing additional clarifications and justifications for their methodological choices and the scope of their study. The reviewers were generally satisfied with the authors' responses.

**Additional Comments On Reviewer Discussion:**

Nil.

---

### Decision · Program_Chairs · 2025-01-22

Accept (Poster)